# Stem Cells in Kidney Ischemia: From Inflammation and Fibrosis to Renal Tissue Regeneration

**DOI:** 10.3390/ijms24054631

**Published:** 2023-02-27

**Authors:** Rosario Cianci, Mariadelina Simeoni, Eleonora Cianci, Oriana De Marco, Antonio Pisani, Claudio Ferri, Antonietta Gigante

**Affiliations:** 1Nephrology Unit, Department of Translational and Precision Medicine, Sapienza University of Rome, 00185 Rome, Italy; 2Department of Translational Medical Sciences, University of Campania ‘Luigi Vanvitelli’, 80131 Naples, Italy; 3Department of Public Health, Nephrology Unit, University of Naples “Federico II”, 80131 Naples, Italy; 4Internal Medicine and Nephrology Unit, Department of Clinical Medicine and Public Health, San Salvatore Hospital, University of L’Aquila, 67100 L’Aquila, Italy

**Keywords:** ischemic nephropathy, renal hypoxia, renal artery stenosis, renal tissue oxygenation, chronic kidney disease, renal staminal cells, NGAL

## Abstract

Ischemic nephropathy consists of progressive renal function loss due to renal hypoxia, inflammation, microvascular rarefaction, and fibrosis. We provide a literature review focused on kidney hypoperfusion-dependent inflammation and its influence on renal tissue’s ability to self-regenerate. Moreover, an overview of the advances in regenerative therapy with mesenchymal stem cell (MSC) infusion is provided. Based on our search, we can point out the following conclusions: 1. endovascular reperfusion is the gold-standard therapy for RAS, but its success mostly depends on treatment timeliness and a preserved downstream vascular bed; 2. anti-RAAS drugs, SGLT2 inhibitors, and/or anti-endothelin agents are especially recommended for patients with renal ischemia who are not eligible for endovascular reperfusion for slowing renal damage progression; 3. TGF-β, MCP-1, VEGF, and NGAL assays, along with BOLD MRI, should be extended in clinical practice and applied to a pre- and post-revascularization protocols; 4. MSC infusion appears effective in renal regeneration and could represent a revolutionary treatment for patients with fibrotic evolution of renal ischemia.

## 1. Introduction

Ischemic nephropathy is a condition characterized by progressive renal function loss, mainly due to atherosclerotic disease of the great renal vessels. It is one of the leading causes of chronic kidney disease (CKD) and dialysis initiation. Consequently, this condition is associated with increased hospitalization and mortality rates and worsened quality of life [1].

Occlusive and/or sub-occlusive atherosclerosis of the renal artery and/or its pre- and intrarenal branches [2,3] are the most common features of ischemic nephropathy. Conversely, this condition rarely depends on renal artery fibrodysplasia in young patients [4]. The span of severity of renal ischemia varies in relation to the extension of intrarenal damage and vascular rarefaction. Several studies indicate that chronic renal vascular obstruction can trigger a pathological cascade characterized by hypoxia, inflammation, microvascular rarefaction, and fibrosis [5]. Fibrotic parenchymal substitution occurs in the long-term and is an irreversible condition that questions the success of revascularization procedures [6]. To date, only a few studies have examined hypoxia-related inflammation biomarkers and their changes after percutaneous transluminal renal angioplasty (PTRA) with and without stenting [7,8]. A more well-explored focus is that on the effects of different reno-protective drugs on pro-inflammatory and pro-fibrotic molecular pathways in chronic renal ischemia; however, we still see an increase worldwide of patients with atherosclerotic ischemic nephropathy that initiate dialysis [9,10].

Current knowledge on ischemia-dependent renal damage pathways suggests additional therapeutic approaches beyond reno-protective drugs and kidney revascularization, including cell regeneration, mitochondrial protection, and/or angiogenic cytokine therapy. Renal function preservation and/or restoration in ischemic nephropathy now appears possible through the stimulation or infusion of mesenchymal stem cells (MSCs) [3,7].

In this review, we provide an overview of two main aspects of ischemic nephropathy: the role of hypoxia on inflammation, fibrosis, and vascular rarefaction, and the emerging field of MSC therapy.

## 2. Renal Hypoxia

In most renal diseases, tissue hypoxia is associated with vascular damage, the so-named nephroangiosclerosis (NAS) [6,11,12]. The correlation between reduced renal blood flow (RBF) and parenchymal oxidative and inflammatory damage in renal atherosclerosis is still poorly understood. Some authors suggest hypoxia as the trigger of a “common pathway” leading to renal fibrosis substitution after tubulo-glomerular damage [2,5]. This depends on the enhanced oxygen consumption by the kidney with the adaptative increase in RBF. As opposed to other organs where the increase in blood flow is followed by a reduction of the hypoxic state, the kidney has the ability to overcome hypoperfusion, but at the high cost of oxidative stress [13]. Several factors (basal cell oxygen consumption, nitric oxide, angiotensin II, etc.) are involved, and they trigger important intracellular pro-inflammatory and pro-fibrotic cascades that are at the basis of renal damage progression [13,14].

NAS is characterized by chronic hypoxia that contributes to both tubulointerstitial fibrosis and glomerulosclerosis [15]. Neusser et al., aimed to analyze the genome-wide expression data from micro dissected glomeruli to evaluate the role of hypoxia in NAS glomerulosclerosis. They found a great number of genes whose expressions are regulated by hypoxia-inducible factors (HIFs). This evidence appears quite interesting in relation to the shared histological features observed in glomerular sections of NAS patients. In particular, the immune–histological analysis evidenced that chemokine C-X-C motif receptor 4 (CXCR4) and 12 (CXCR12) have pivotal roles in induced hypoxia, leading to both tubulointerstitial fibrosis and glomerulosclerosis in NAS [16].

It is well-known that hypoxia is a landmark common to all cardiovascular diseases (hypertension, arterial aneurysms, atherosclerosis, pulmonary arterial hypertension, congestive heart failure, etc.), and HIFs are involved in many mechanisms such as erythropoiesis, angiogenesis, and inflammation [17]. Any therapeutic approach addressed at stabilizing HIF activity might have a potential benefit in ischemic nephropathy, especially in patients with NAS; however, this important question has not been addressed and/or demonstrated in any trial so far.

Sami Palomäki et al., reported an upregulation of HIF-1α in MSCs [18]. This could explain how the increase in intrarenal HIF-1α may stimulate angiogenesis and increase blood flow, as reported by Jennifer E Ziello et al. [19]. Many genes not only involved in angiogenesis but also in inflammation, such as vascular endothelial growth factor (VEGF), matrix metalloproteinase 2 (MMP2), cathepsin D (CATHD), keratin (KRT), and tumor growth factor-β (TGF-β), are targets of the HIF-1 transcription complex, a controversial factor in hypoxic renal damage progression. Jiacheng Sun et al., tested the proangiogenic activity of exosomes obtained from HIF-1α-modified MSCs in myocardial infarction. They demonstrated the relevant capacity of these MSC-derived exosomes in angiogenesis, confirming the therapeutic potential of these engineered cells [20]. The novel drug class of HIF stabilizers recently approved for the treatment of secondary anemia in CKD patients should be investigated for possible pleiotropic effects on all inflammation, angiogenesis, and MSC activity.

Transition to CKD in revascularized patients with renal artery stenosis (RAS) depends on the coexistence of extensive intrarenal NAS with reduced microvascular density [2]. According to the chronic hypoxia hypothesis, the loss of glomeruli also induces hypoperfusion of peritubular capillaries leading to tubular damage and vascular rarefaction. Diffuse intrarenal atherosclerosis is the main feature of a condition called atherosclerotic renovascular disease (ARVD), in which endovascular revascularizing procedures would not be indicated and are associated with worse outcomes. In ARVD, inflammation and fibrosis are especially evident and explain the typical and untreatable evolution to CKD [12].

As reported by several authors, in ischemic nephropathy, the rarefaction of the vascular bed is always present, but with variable severity. In fact, a stable 30–40% reduction in renal blood flow (RBF) is associated with unchanged intrarenal oxygenation after PTRA [5,7]. This partly depends on compensation due to altered shunting. Moreover, a lower RBF leads to GFR decline with diminished renal oxygen consumption, and the kidney varies the state of oxygenation between the cortex and the medulla [21].

As the reduction in RBF increases, the transition from a purely hemodynamic phase to an inflammatory one begins, since the reduction in renal oxygenation will activate the proinflammatory and fibrotic pathways, which could be reversed if renal ischemia is promptly treated [22,23,24]. In agreement with relevant studies, inflammation plays a key role in the transition from acute kidney injury (AKI) to CKD in ischemic nephropathy [25,26,27]. In hypoxic renal tissue, macrophagic infiltrates and damaged tubular cells release several inflammatory and profibrotic cytokines and chemokines such as TGF-β, VEGF, monocyte chemoattractant protein-1 (MCP-1), and neutrophil gelatinase-associated lipocalin (NGAL) [27,28]. TGF-β is one of the major actors, since it induces myofibroblast proliferation, which leads to the inhibition of both proteinases production and matrix degradation [29]. Several reports indicate that most inflammation mediators in ischemic nephropathy are induced through the activation of renin–angiotensin–aldosterone system (RAAS), whose up-regulation is aimed at compensating hypoperfusion and is mostly accompanied by hypertension and proteinuria [30,31,32]. This underlines the importance of the use of anti-RAAS agents in ischemic patients not eligible for revascularization [3,33].

### 2.1. How to Evaluate and Quantify the Hypoxic State and Inflammatory Cell Activity

Several methods to measure intrarenal oxygen tension are available. In studies conducted on animals, the use of polarographic oxygen microelectrodes has provided most of the current information on kidney oxygenation [34,35,36]; however, anesthetic agents used during experiments could impair the quality of results [37,38]. In this regard, at least three solutions have been proposed: (1) the use of implantable microelectrodes in combination with telemetry devices [39]; (2) the use of fluorescence optodes [39]; and (3) the use of total intravenous instead of volatile anesthesia [37]. All of these methods have been tested by the Evans group over the last three decades in different experimental settings. In the comparison of microelectrodes and fluorescence, it appeared that both presented some limitations, and their abilities regarding the assessment of renal oxygenation were not perfect. The Evans group, in a recent review, asserted that the real renal oxygenation level comprised those detected using the two techniques. Total intravenous anesthesia, instead, failed to provide stabilization of intrarenal oxygenation in sheep undergoing cardiopulmonary bypass but seemed more effective than volatile anesthesia outside of such invasive surgery [37].

In vivo and in vitro studies have demonstrated an increase in cellular levels of HIFs in ischemic nephropathy [40]. HIF-1α and HIF-2α are two oxygen-sensitive transcription factors acting as mediators of the cellular adaptive response to hypoxia. Consequently, the evidence of an intracellular accumulation of HIFs and an increase in the expression of HIF target genes could be interpreted as evidence of renal tissue hypoxia [41].

Blood NGAL has been proposed as a biomarker of renal ischemia, and it is a reliable and early biomarker of AKI [42,43]. Textor et al., reported that NGAL levels are inversely correlated with eGFR in patients with ischemic nephropathy [44]. Moreover, our group has recently published original data on NGAL and circulating renal stem cell (RSC) changes before and after PTRA and stenting in five patients with RAS. We reported that the presence of high NGAL levels before revascularization is associated with lower RSC expression. However, an adaptative and reparative increase in the latter was also observed independent of baseline NGAL levels [8].

The functional assessment of renal tissue oxygenation in humans was possible only after the introduction of blood oxygen level-dependent magnetic resonance imaging (BOLD-MRI) [12,45]. This technique allows for the study of renal oxygenation status in relation to the degree of mild (G1), moderate (G2), and severe (G3) RAS. BOLD-MRI sensitivity and specificity in the identification of renal ischemia are related to the paramagnetic quality of deoxygenated hemoglobin, which modifies magnetic realignment after a field interruption, as opposed to to oxygenated hemoglobin, which has no active magnetic component [45,46] (Figure 1). As for vascular rarefaction, it might be due to the partial volume effect that also leads to a lower amount of deoxyhemoglobin per pixel with possible inaccuracy of its detection using BOLD-MRI.

### 2.2. Role of Mesenchymal and Resident Renal Stem Cells in Kidney Inflammation and Repair

Several renal damage mechanisms share common ground in kidney ischemic insult and lead to fibrosis and renal function loss [47]. Considering the high economic and social costs of renal replacement therapies, this represents a major health problem worldwide [48]. Similar to the nervous system and the heart, the kidneys do not appear to be able to effectively repair themselves after acute or chronic damage. Many reports on stem cells in the last two decades have opened up a novel and hopeful prospect, that of the renal regeneration [2,5,7,49].

Mesenchymal stem cells (MSCs) are fibroblast-like stromal cells that have been isolated first from the bone marrow and successively from several other organs and tissues, including the kidney [50,51,52]. Many recent studies have focused on the exploration of their therapeutic potential in different renal damage conditions. MSCs ability to migrate and/or differentiate in the kidney following several renal insulting mechanisms defines the extraordinary repairing potentiality of these peculiar cells. Once migrated or activated in the damaged kidney, MSCs produce and release, through extracellular vesicles and exosomes, a variety of cytokines and chemokines able to reduce inflammation and increase different reparative pathways. Paracrine and/or endocrine mechanisms are involved, with effects on both immune response modulation and cellular replacement [53,54].

In mammalian kidneys, unlike those of cartilaginous fish, MSC proliferation and differentiation ceases after birth [55]. As a consequence, the response of mammalian kidneys to irreversible renal mass loss following a partial or total nephrectomy consists of compensatory hypertrophy of the adjacent nephrons or of the contralateral kidney, respectively [56,57]. Conversely, in the case of AKI, the adult kidney has some ability to repair and remodel, which can ultimately lead to restoration of renal structure and function [58]. Looking to renal ischemia in all its variants (RAS, RAS+NAS, and ARVD) leading to both tubular epithelial cell and podocyte death [3,8], it is known that the damaged tubule will undergo repair over a period of 8–12 weeks, which is most likely mediated by special cells previously classified as survivor epithelial cells. These are expressed close to the damaged renal tissue and correspond to RSCs [59,60]. These cells are thought to be MSCs renally committed, migrated, and differentiated to RSCs at the site of tissue damage in response to different stimuli. It has been reported, in fact, that MSCs express several cytokine and chemokine receptors that may be functional during migration to the sites of inflammation [61,62]. In addition, it has been reported that in the course of ischemia, HIF-1α is able to protect MSCs against oxygen–glucose deprivation-induced injury through both the induction of autophagy and the activation of the intracellular PI3K/AKT/mTOR signaling cascade [63]. Under hypoxic insult, the expression of stromal cell-derived chemokine factor 1 (SDF-1) has been found to be upregulated within the kidney [64]. The finding of SDF-1 receptor (CXCR4) expression on MSCs is a further demonstration of the chemiotactic activity elicited by hypoxia on MSCs, with a protective and reparative aim [65]. It has been reported, in support of this, that the pre-incubation of MSCs with TNF-α is followed by increased MSC migratory capacity. This indicates that the SDF-1/CXCR4 interaction may mediate the localization of exogenously injected MSCs to hypoxically damaged renal tissue with potential therapeutic effect [66,67].

The mechanisms preserving the integrity of MSCs and addressing their migration to damaged tissue increase the hope for future successful use of these cells in the treatment of renal ischemia, even in the case of extensive intrarenal vascular-bed compromise and fibrosis onset.

Resident RSCs have been found in the parietal epithelium of the Bowman capsule at the urinary pole of the glomerulus in association with different kidney insults [68,69,70]. Their identification and localization were based on studies of RSCs’ specific surface antigenic expression. Specifically, the finding of CD24, CD133 [70,71], and podocalyxin expression [72] in some parietal cells of the glomerular capsule, next to the tubule origin in animal models of AKI, demonstrated the existence of RSCs and their ability not only to replace tubular cells but also to regenerate podocytes after injury [59,65]. Of note, we have recently reported evidence of circulating RSCs showing similar surface antigen profiles to resident cells. Also, circulating RSCs showed the ability to repair kidneys after ischemia/reperfusion damage [8,30]. Several authors have also reported stem cell-dependent immunomodulation able to reduce the intrarenal inflammatory response invariably associated to any kidney insult [44,73,74,75]. The secretion of paracrine factors committed to inhibiting the release of interleukin-1α (IL-1α) and tumor necrosis factor α (TNF-α) and to increasing the secretion of the IL-1 receptor antagonist are reported mechanisms explaining the ability of differentiated MSCs to repair injured kidneys [76,77]. In addition, it has been demonstrated in rodents that these cells are able to secrete several growth factors with different biological effects ranging from angiogenesis to mitogenesis, cell proliferation, and apoptosis prevention [78,79,80,81,82,83,84]. Moreover, it has been reported that differentiated MSCs are also able to influence the activity of macrophages by enhancing their pivotal role in cell replacement and repair through the reduction of inflammation, the clearing of apoptotic cells and debris, and the promotion of tissue remodeling and regeneration [85,86]. Macrophage activation and function depend on the inflammatory stimuli received by the damaged tissue microenvironment. As the repair process progresses from the initial inflammatory phase to the remodeling phase, macrophages successively exhibit variable polarization and activation states [87].

### 2.3. Mesenchymal and Renal Stem Cell Use in Renal Ischemia

Inflammatory damage and fibrosis in the ischemic kidney play the main negative roles in PTRA failure. They are associated with worse outcomes since they induce renal damage irreversibility. Beyond revascularization, anti-RAAS agents and novel molecules such as SGLT2 inhibitors and anti-endothelin agents are most often recommended to prevent the fibrotic evolution of renal damage [88,89,90]. However, no clear evidence for renal tissue revitalization has been reported in association with the use of these drugs [7,91,92]. Regenerative therapies appear to be the most desirable and future solution to this major CKD cause. The prospect is that of a substantial change in outcomes using stem cells in ischemic CKD patients, with possible dialysis retardation or avoidance (Figure 2).

The presence of MSCs in different tissues, where they show the ability to differentiate into resident mature cells, continues throughout life; however, it has been reported that the number and the functionality of these cells can be influenced by age [93]. In embryos, the differentiation status of totipotent zygotic cells stops at the stage of pluripotent embryonic cells, which constitute the germ layer. After this phase, stem cells will reside in specific areas of several tissues, where they can move from quiescence to activation at any moment. They can respond to specific triggers with differentiation to mature cells with consequent tissue regeneration. This typically occurs in the concomitance of renal inflammation and fibroblast activation [94,95]. On the other hand, it has been reported that MSCs collected from patients with ARVD might exhibit impaired function, senescence, and DNA damage, possibly determined via epigenetic mechanisms.

In a recent review by Yanmeng Yang et al., the authors concluded that a hypoxic environment can positively influence different functions (proliferation, survival, homing, differentiation, and paracrine activities) of MSCs. As a consequence, the authors propose hypoxia preconditioning (HPC) of MSCs as an engineering approach that could potentially improve the therapeutic efficacy of these cells [96]. In fact, as described by Isik et al., in cultures of MSCs collected from pigs with RAS, HPC exerted beneficial effects on cell viability in terms of proliferation and differentiation [97]. Moreover, the downregulation of chromatin-modifying enzymes resulted in a reduction of MSC senescence and dysfunction. Therefore, the authors concluded that HPC of MSCs can be considered as a methodological optimization able to improve autologous cell therapy in patients with ischemic nephropathy [75]. In vivo experiments on rats with ischemia-reperfusion injuries treated with HPC MSCc versus non-preconditioned MSCs have demonstrated an enhanced ability to reduce profibrotic circulating factors and to induce VEGF-related angiogenesis in rats with ischemic nephropathy [98].

Some reports in animal models have explored the possible restorative role of MSCs after renal reperfusion and have shown encouraging and promising results [27,33]. One of these emerged in the preclinical studies conducted by Textor’s group [44]. Renal microcirculation reparation, inflammatory damage attenuation, and renal function restoration were the observed effects of MSC infusion. These benefits seemed to be mediated by paracrine effects linked to the release of soluble mediators and extracellular vesicles, and to the activation of pericytes.

In an interesting study on an RAS pig model, Chen XJ et al., also reported that low-energy shock wave (SW) therapy and the administration of fat-derived MSCs can stimulate both the repair of ischemic renal damage and the regeneration of renal vasculature. Specifically, researchers administered fat-derived MSCs directly into the renal artery to enhance SW therapeutic power and preserve the function and structure of the stenotic kidney, and they obtained these important signs of renal revitalization [99].

Xin Zhang et al., conducted similar experiments in twenty-two pigs with RAS that were treated with SW therapy and compared the effects of MSCs infusion to those observed in uninfused controls. Four weeks after treatment, individual RBF, GFR, and oxygenation were assessed in vivo, while renal microcirculation, fibrosis, and oxidative stress were studied ex vivo. Based on the results, the authors reported that the intraarterial infusion of MSCs enhanced the pro-angiogenic effect of SW treatment, which resulted in an improvement of both RBF and GFR. They concluded that SW therapy is fundamental for ameliorating renal microvascular aspects, but MSCs are necessary for rebuilding the vascular structure [100]. A recent study by the same group revealed the reasons for their previous results. They demonstrated, in fact, that low-energy SW treatment can promote the homing of endothelial cell progenitors to the stenotic kidney [101]. Translating this evidence to humans, Sune Moeller Skov-Jeppesen et al., recently tested the combination of SW therapy and fat-derived MSC administration in 14 patients with diabetes mellitus and Stage 3 CKD. All in-study patients showed a good tolerance to the treatment, although GFR and albuminuria did not significantly change in the follow-up. More studies in larger populations would be needed to achieve sufficiently high statistical power.

In different interesting studies conducted on animals by the Fan Yang group and others, fat-derived MSCs showed a marked reparative, anti-inflammatory, and anti-fibrotic ability in streptozotocin-induced diabetic nephropathy [102,103,104]. Abumoawad A et al., tried the same protocol in patients with ARVD through the administration of different doses of fat-derived autologous MSCs [7]. Specifically, thirty-nine subjects with ARVD were studied and selected for intraarterial infusion of MSCs. Of these, only twenty-one subjects were enrolled and divided into three groups with different MSC dose assignations: 1, 2.5, and 5.0 × 105 cells/kg, respectively. Six weeks prior to the scheduled MSC administration, approximately 2 g of subcutaneous abdominal fat was biopsied from each patient and frozen. Four to six days prior to infusion, MSCs were thawed and cultured, and subsequently infused into the renal artery. Only with the greater dose of MSC infusion (5.0 × 105 cells/kg) did patients show increased tissue oxygenation with blood pressure and renal function improvement. Furthermore, inflammatory-state regression was confirmed through the reduction of NGAL and MCP-1 levels. At the 3-month follow-up, a stable improvement in renal oxygenation was finally confirmed via BOLD-MRI findings [7].

Podocytes and tubular cell progenitors have been found in the Bowman’s capsule. Both cell typologies show a heterogeneous ability to differentiate and regenerate after renal ischemia/reperfusion [105,106]. Our group previously reported the first human case of successful post-reperfusion renal recovery accompanied by a significant increase in circulating RSCs in a revascularized patient with bilateral RAS [22].

Moreover, in our recent report on five patients with renovascular hypertension and renal ischemia due to atherosclerotic RAS, we evaluated all renal-resistive indexes (RRI), circulating RSCs, and NGAL at baseline and at different timepoints after PTRA. We were able to confirm that PTRA can produce different outcomes in relation to the degree of downstream vascular bed compromise. Based on our protocol, we were able to observe that the modest basal activation of RSCs in some cases was due to hypoxia, which was closely correlated with high NGAL levels and elevated RRI [8]. We hypothesized that in patients with worse outcomes, the compromised vascular bed associated with high levels of NGAL enabled RSC migration into the renal ischemic tissue or did not permit the survival and differentiation of resident RSCs. Conversely, our observation showed that in cases with better outcomes, adequate levels of RSCs at the baseline were linked to the absence of NAS and lower levels of NGAL. The latter were an expression of angiogenesis and tubular cell repair that led to the differentiation of RSCs into podocytes or tubular cells with functional recovery after PTRA [8].

## 3. Conclusions

Our review, mainly focused on the mechanisms of tissue damage and regeneration in renal ischemia, further underlined the importance of an adequate organ blood supply for the integrity of kidney function. In the last decade, compared to medical therapy alone, PTRA with/without stenting has not shown any therapeutic superiority in the treatment of renovascular disease; however, it remains the main therapy for RAS in patients showing an adequately preserved downstream vascular bed.

The use of anti-RAAS drugs, SGLT2 inhibitors, and/or anti-endothelin agents is also recommended, but beyond a “point of no return”, ischemic renal damage becomes irreversible. Ischemic nephropathy is a set of morbid conditions, distinct but connected to each other, and all dependent on the reduced perfusion that activates several harmful pathological processes mostly characterized by inflammation and fibrosis. Recent studies suggest that the adaptability of kidneys to moderate RBF reduction is transiently well tolerated, and that PTRA should be performed only after severe inflammation and fibrosis have been excluded. To this purpose, the assay of NGAL and other biomarkers along with BOLD-MRI should be extended in clinical practice and possibly applied to pre- and post-revascularization protocols; however, their usefulness in the decision-making process needs to be confirmed and validated in dedicated trials.

The interest on MSCs has rapidly increased in recent years. Their preliminary use with a renal regenerative, anti-inflammatory, and anti-fibrotic aim in several studies conducted on cell cultures, animals, and humans has shown promising translational results. More studies should also be performed to increase understanding how MSC senescence could be overcome to enhance the therapeutic potential of these cells. HPC of MSCs and low-power SW therapy are effective engineering proposals for optimizing MSCs’ therapeutic potential. Even MSCs’ extracellular vesicles show promising application in the treatment of hypoxia-related renal damage.

Further translational research studies that combine all hypoxic statuses, inflammation degrees, fibrotic pathways, and roles of MSCs in renal regeneration would be essential for a more accurate selection of patients with ischemic nephropathy eligible for revascularization. The final aim is that of timely and effective treatment. MSCs infusion, used alone or combined with PTRA and reno-protective drugs, could represent an additional future innovative treatment for patients with any form of renal ischemia.

## Figures and Tables

**Figure 1 ijms-24-04631-f001:**
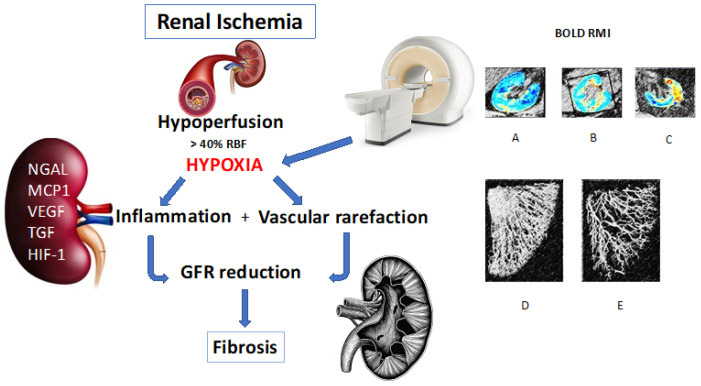
This figure depicts the intrarenal pathogenic cascade triggered by kidney ischemia. Each stage of renal response to ischemia is detected using specific diagnostic tools. BOLD-MRI is sensitive in identifying critical organ perfusion, hypoxia, and possibly vascular rarefaction; NGAL, MCP-1, VEGF, and TGF-β are early biomarkers of intrarenal inflammation; in the long-term, GFR decline is indicative of fibrotic renal involution. (A) Normally oxygenated kidney; (B) Initial kidney deoxygenation; (C) Critically reduced kidney oxygenation; (D) Normal renal microvascularization; (E) Vascular rarefaction in an ischemic kidney.

**Figure 2 ijms-24-04631-f002:**
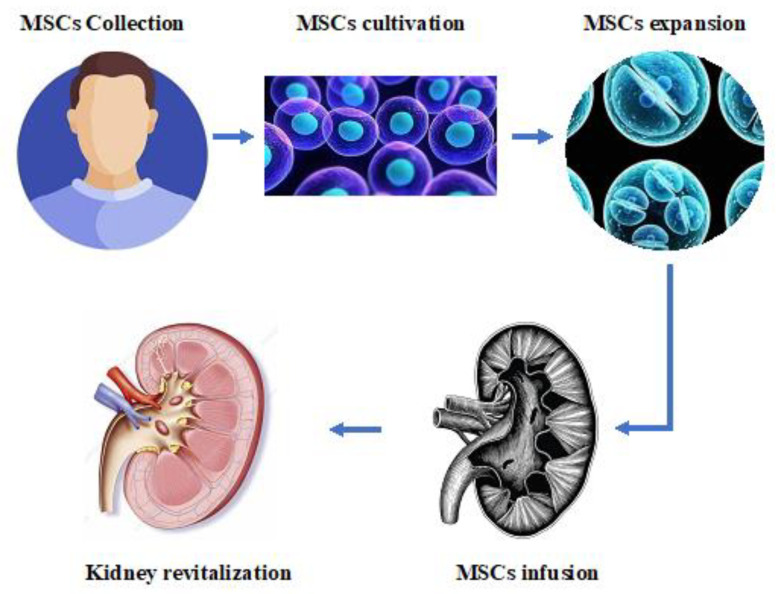
This figure depicts the experimental setting of renal revascularization using MSC infusion.

## Data Availability

No new data were created or analyzed in this study. Data sharing is not applicable to this article.

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
