# Peer review of "Stem Cells in Kidney Ischemia: From Inflammation and Fibrosis to Renal Tissue Regeneration"

_ijms, 2023, doi:10.3390/ijms24054631_

Round 1

Reviewer 1 Report

This is a nicely written review by a well-established Italian researchers about renal artery stenosis, stem cells and ischemic nephropathy, a subject that has fascinated many physiologists, radiologists, internists, nephrologists and other clinicians in recent years. The review summarizes two main axes: the role of hypoxia on inflammation, fibrosis and vascular rarefaction, and the emerging field of MSC therapy.

Some parts need to my opinion revision before the paper can be accepted for publication. First of all, the role of HIF in fibrosis and ischemia is a two-edged sword, as HIF does not only lead to the production of VEGF, EPO and  MMP-2 (line 108), but can also stimulate TGFb, a well-known pro-fibrotic factor. In this respect, whether Prolyl hydroxylase inhibitors and other HIF stabilizers  will have anti-or pro-fibrotic properties is, to the best of my opinion, unknown yet.

Secondly, the authors state that renal ischemia occurs when RAS is larger than 40%, and cite a paper from 1993 to support their statement. Textor’s group has repeatedly shown, using BOLD-MRI, that renal oxygenation is well maintained till even severe stenosis, well above 40% (for example doi: 10.1016/j.kint.2018.11.039 or PMID: 23917027. This is partly because of compensation thanks to altered shunting, and also because a lower RBF leads to a lower GFR, and thus diminishes renal oxygen consumption (see also some of Fredrik Palms work). I invite the researchers to discuss these findings.   The connection between Hypoxia  and vascular rarefaction (Figure 1) also needs a bit more explanation. In the same figure, the authors mention that GFR reduction leads to fibrosis. This is indeed the case, as loss of glomeruli according to the chronic hypoxia hypothesis also means loss of the perfusion of peritubular capillaries. However, the chronic hypoxia hypothesis is mentioned, but not explained in the text.

Finally, stating that PTRA should be performed only after severe inflammation and fibrosis have been excluded (line 387)is theoretically interesting and probably correct, but the usefulness of NGAL or BOLD MRI in clinical practice and decision making still has to be demonstrated in clinical trials.

In the conclusion, it is a bit strange to state that PTRA is the gold standard therapy, as large randomized controlled trials did not convincingly demonstrate a benefit of PTRA over medical therapy. I agree that this is possibly linked to problems with patient selection, but I advise the authors to downsize this statement.

Minor remarks:

Introduction

-Sentences tend to be too long (for ex the first sentence of the introduction), and some English revision would be welcome

-Please include a definiton of ischemc nephropathy and ARAS.

-The aim is not very clear. Instead of stating “analyzing the single frame of the kidney hypoperfusion-dependent inflammation”, it would be better to state that they provide an overview of ..

-line 166: why say that ‘Even Textor et al reported that..’. Was his team so much against the use of NGAL?

-Figure 1: BOLD MRI is indeed sensitive to identifying critical organ oxygenation, but vascular rarefaction may due to the partial volume effect also lead to a lower amount of deoxyHb per pixel, and give the impression of increased oxygenation.

Reviewer 2 Report

The review:” Stem cells in kidney ischemia: from inflammation and fibrosis to renal tissue regeneration” is focused on kidney hypoperfusion-dependent inflammation and its influence on the renal tissue ability to self-regeneration. Moreover the authors provided an overview of the advances in the regenerative therapy with  mesenchymal stem cells (MSCs) infusion.

I personally think that systematic-review is a more powerful tool than classical review and scientific community must be encouraged to this new way of performing a review work. Anyway, the present review is well structured, the bibliography is accurate and up to date. I have only one suggestion to the authors to check carefully the editing.
